# Spin Crossover and Thermochromism in Iron(II) Complexes with 2,6-Bis(1*H*-imidazol-2-yl)-4-methoxypyridine

**DOI:** 10.3390/ijms24129853

**Published:** 2023-06-07

**Authors:** Olga G. Shakirova, Irina A. Os’kina, Evgeniy V. Korotaev, Sergey A. Petrov, Natalia V. Kuratieva, Alexsei Ya. Tikhonov, Lyudmila G. Lavrenova

**Affiliations:** 1Nikolaev Institute of Inorganic Chemistry, Siberian Branch, Russian Academy of Sciences, 630090 Novosibirsk, Russia; shakirova_olga@mail.ru (O.G.S.); korotaev@niic.nsc.ru (E.V.K.); kuratieva@gmail.com (N.V.K.); 2Department of Chemistry and Chemical Technologies, Faculty of Machinery and Chemical Technologies, Federal State Budget Institution of Higher Education, Komsomolsk-na-Amure State University, 681013 Komsomolsk-on-Amur, Russia; 3N.N. Vorozhtsov Novosibirsk Institute of Organic Chemistry, Siberian Branch, Russian Academy of Sciences, 630090 Novosibirsk, Russia; oi@nioch.nsc.ru (I.A.O.); alyatikh@nioch.nsc.ru (A.Y.T.); 4Institute of Solid State Chemistry, Siberian Branch, Russian Academy of Sciences, 630128 Novosibirsk, Russia; petrov@solid.nsc.ru

**Keywords:** synthesis, complexes, iron(II), 2,6-bis(1*H*-imidazol-2-yl)-4-methoxypyridine, spectroscopy, spin crossover

## Abstract

New iron(II) complexes with 2,6-bis(1*H*-imidazol-2-yl)-4-methoxypyridine (**L**) of the composition [FeL_2_]A_n_∙mH_2_O (A = SO_4_^2−^, n = 1, m = 2 (**I**); A = ReO_4_^−^, n = 2, m = 1 (**II**); A = Br^−^, n = 2, m = 2 (**III**)) have been synthesized and investigated. To determine the coordination ability of the ligand, a single crystal of a copper(II) complex of the composition [CuLCl_2_] (**IV**) was obtained and studied by X-ray technique. Compounds **I**–**III** were studied using methods of X-ray phase analysis, electron (diffuse reflection spectra), infrared and Mössbauer spectroscopy, static magnetic susceptibility. The study of the µ_eff_(T) dependence showed that the ^1^A_1_ ↔ ^5^T_2_ spin crossover manifests itself in the compounds. The spin crossover is accompanied by thermochromism: there is a distinct color change orange ↔ red-violet.

## 1. Introduction

In coordination compounds of metals with the electronic configuration *d*^4^_-_*d*^7^, at a certain strength of the ligand field, the phenomenon of spin crossover (**SCO**) manifests itself, is a change in spin multiplicity under the influence of external conditions, such as temperature, pressure, irradiation with light of a certain wavelength, external magnetic or electric fields, light-controlled ligand isomerization and solvation/desolvation [1,2,3,4,5]. Coordination compounds of iron(II) with poly-nitrogen-containing ligands are of particular interest due to the fact that in many of them SCO is accompanied by thermochromism, is a reversible color change at the spin transition temperature. Bistable molecular sensors can be in demand for a wide range of applications, including in the field of nanotechnology, such as display and memory devices, sensors [6], MRI contrast agents [7], thermoelectrochemical cells [8], etc.

Spin crossover in iron(II) complexes, in terms ^1^A_1_ (S = 0, low-spin, **LS**) ^5^T_2_ (S = 2, high-spin, **HS**), is always unique for each iron(II) complex. The transition can be abrupt or gradual, complete or incomplete, may have hysteresis or not on the _eff_(T) dependence curve, have one or two stages. The temperatures of the direct transition (when heated, **T_c_**) and the reverse ones (when cooled, **T_c_**) depend on the composition of the compounds and vary widely. The fact that the low-spin state is diamagnetic makes it possible to estimate the temperature-independent paramagnetism of John Hasbrouck van Vleck.

At present, our research team synthesizes and investigates a number of iron(II) complexes with derivatives of 2,6-bis(1*H*-imidazol-2-yl)pyridine (**L***) and various external anions of the composition [FeL*_2_]A_n_∙mH_2_O (n = 1, 2; m = 0–2). Both the literature data [9], and the results of X-ray and EXAFS spectroscopy obtained by us show that ligands of this class are coordinated to iron(II) in a tridentate-cyclic (pincer) type by an N atom of pyridine and two N(3) atoms of imidazole cycles. For most of the studied iron(II) complexes with 2,6-bis(1*H*-imidazol-2-yl)pyridine derivatives, a high-temperature one- or two-stage SCO ^1^A_1_ ^5^T_2_ is observed [10,11,12,13,14,15]. In contrast to the iron(II) complexes with 1,2,4-triazoles and tris(pyrazol-1-yl)methanes previously studied by us [5,16,17], we have not observed the phenomenon of thermochromism in a series of Fe(II) compounds with 2,6-bis(1*H*-imidazol-2-yl)pyridine. It can be assumed that the reason for this is the intense violet color of the low-spin form, which even with its low content masks the white color of the high-spin form of complexes at high temperatures. In order to study the effect of the substituent position in the ligand on the characteristics of the SCO system, it seemed advisable to continue studying iron(II) complexes with a new derivative of this class, in which the substituent is not in the imidazole, but in the pyridine cycle. For this purpose, we synthesized a new functionalized ligand with an auxochromic group 2,6-bis(1*H*-imidazol-2-yl)-4-methoxypyridine (**L**, Figure 1).

This work is devoted to the development of a method for the synthesis of a new organic compound 2,6-bis(1*H*-imidazol-2-yl)-4-methoxypyridine, the production of iron(II) complexes based on it and the study of their physicochemical and, in particular, magnetic properties.

## 2. Results

### 2.1. Synthesis and Characterization

We used a convenient synthetic route to obtain 2,6-bis(1*H*-imidazol-2-yl)-4-methoxypyridine **L**. Chelidamic acid was successively converted into the corresponding methyl ester **2**, amide **3** and nitrile **4** using the following reactions [18,19] (Figure 1). Dimethyl 4-hydroxypyridine-2,6-dicarboxylate **1** and dimethyl 4-methoxypyridine-2,6-dicarboxylate **2** were synthesized by the reflux of chelidamic acid hydrate in MeOH with conc. H_2_SO_4_. However, the incomplete conversion of chelidamic acid to dimethyl 4-methoxypyridine-2,6-dicarboxylate **2** as observed under these conditions. Additional amount of dimethyl 4-methoxypyridine-2,6-dicarboxylate **2** was prepared by the alkylation of dimethyl 4-hydroxypyridine-2,6-dicarboxylate **1** with CH_3_I in the presence of K_2_CO_3_ under heating at 50 °C in DMF. 4-Methoxypyridine-2,6-dicarboxamide **3** was isolated in the reaction of dimethyl 4-methoxypyridine-2,6-dicarboxylate **2** with NH_4_OH in refluxing methanol. The treatment of 4-methoxypyridine-2,6-dicarboxamide **3** with trifluoroacetic anhydride in THF at 0 °C in the presence of Et_3_N produced of 4-methoxypyridine-2,6-dicarbonitrile **4**. For the construction of the 1*H*-imidazole, moiety the nitrile **4** was reacted with amino acetaldehyde diethyl acetal [20] leading to the isolation of 2,6-bis(1*H*-imidazol-2-yl)-4-methoxypyridine **L** (Figure 1). The compounds **1**-**4** and the ligand **L** NMR spectra data are given in the Appendix A, Appendix A.

Complexes [FeL_2_]SO_4_∙2H_2_O (**I**), [FeL_2_](ReO_4_)_2_∙H_2_O (**II**), [FeL_2_]Br_2_∙2H_2_O (**III**) and [CuLCl_2_] (**IV**) were obtained from acidified aqueous–ethanol solutions at a molar ratio M:L = 1:2. Elemental analysis of the obtained phases **I**–**III** showed that the phases have the appropriate composition. The resulting iron(II) complexes are slightly soluble in water and ethanol. Elemental analysis of the bright green complex **IV** showed that the crystals have a composition of Cu:L = 1:1. This complex is highly soluble in water, ethanol and acetone. All compounds are stable for a long time when stored in the air at room temperature. The thermal behavior of both the ligand and the complexes was studied using TG/DSC measurements. The compounds lose crystallization water in the temperature range of 70–120 °C, remaining stable in an anhydrous state up to 350 °C (Appendix A).

### 2.2. X-ray Structure Determination

X-ray phase analysis data indicate the crystallinity of powdered samples (Appendix A). However, we were unable to grow single crystals of iron(II) complexes suitable for analysis. In order to determine the coordination ability of the new ligand, we obtained and studied a single crystal of a copper(II) chloride complex with **L** of composition [CuLCl_2_].

X-ray diffraction analysis of complex **IV** (Table 1 and Table 2) indicates that the [CuLCl_2_] phase crystallizes in a monoclinic crystal system. The crystal packing is molecular (Figure 1). The coordination environment of the central Cu(II) ions is a slightly distorted square pyramid. **L** is coordinated to copper(II) in a tridentate-cyclic type by the N1 atom of the pyridine and the N2 and N3 atoms of the imidazole cycles. Due to the steric rigidity of chelate cycles, the “central” Cu–N1 distance is 0.05–0.07 Å shorter than the “lateral” ones (thus, 1.983(3) Å and 2.032(3), 2.054(3) Å); the chelate angles N1–Cu–N2 and N1–Cu–N3 are within 78.78(13)–78.94(13)° (Table 1). The ligand molecule **L** has only a slight deviation from planarity: the angle of inclination of the root-mean-square planes of the imidazole fragments with respect to the plane of the pyramidal fragment does not exceed 1°, and the methyl group is deviated by only 4°. The coordination polyhedron of copper(II) is completed into a tetragonal pyramid due to two chloride ions (d_Cu–Cl_ is equal to 2.2137(11) Å in the equatorial plane and 2.6761(11) Å in the axial position). Additionally, the structure is stabilized by stacking interactions between the two nearly planar organic ligands of adjacent molecules with the shift in the equatorial plane and the mean-square distance between the square pyramidal bases of about 3.42(8) Å. Neighboring molecules are located with an offset in the equatorial plane (Figure 2). The centroids of such pseudo-dimers form a two layered close packing motive in the (1 0 1) plane system.

### 2.3. Infrared Spectroscopy

Table 3 shows the main vibrational frequencies (cm^−1^) in the IR spectra of **L** (Appendix A) and complexes **I**–**III** (Appendix A). In the low-frequency region of spectra **I**–**III** (Appendix A) bands appear that can be attributed to Fe^HS^–N valence vibrations.

### 2.4. Diffuse Reflectance Spectroscopy

In the DRS of complexes (Appendix A) in the region of 1000–900 nm, one wide absorption band is observed. Bands with maxima at 936 nm (**I** and **III**) and 923 nm (**II**) can be attributed to the d-d transition ^5^T_2_ → ^5^E in a weak distorted octahedral field of ligands. The position of these bands is characteristic of the spectra of high-spin octahedral iron(II) complexes with nitrogen-containing ligands.

In addition, intense metal-ligand charge transfer bands are observed in the 450–650 nm region ν_1_ (e_g_ → $\pi_{L}^{*}$) (λ_max_ = 433 nm (**I**); 407 nm (**II** and **III**)) and ν_2_ (t_2g_ → $\pi_{L}^{*}$) (λ_max_ = 639 nm (**I**–**III**)).

### 2.5. Mössbauer Spectra

The Mössbauer spectra of complexes **I**–**III** are quadrupole doublets (Figure 3, Table 4), their parameters correspond to the HS state of Fe (II).

### 2.6. Study of the Complexes Magnetic Properties

The temperature dependences of the effective magnetic moment for complexes **I**, **II** and **III**, as well as their dehydrated derivatives **Ia**, **IIa**, **IIIa** (measurements were carried out in a helium atmosphere after removal of crystallization water) are shown in Figure 4, Figure 5 and Figure 6. 

All the studied compounds are in a high-spin state at room temperature and, with the exception of complex **III** (measurements on an air-sealed sample), pass into a low-spin state at T < 250–120 K. It should be noted that the most complete transition is observed for complexes **I** and **Ia**. In this case, the transitions are accompanied by a reversible color change from orange to various shades of red-violet (an example of a color change for complex **Ia** is shown in the box in Figure 4a). The values of the effective magnetic moments of dehydrated complexes **Ia**, **IIa** and **IIIa** in the high-spin state at maximum temperature is 5.14, 5.28 and 4.61 μ_β_, respectively. For the initial complexes **I**–**III**, the values 5.26, 5.35 and 4.73 μ_β_, are observed. The obtained values of µ_eff_ differ from the spin only value of 4.9 μ_β_, however, they range within the experimental values of 4.6–5.7 μ_β_ for Fe (II) [21,22]. For complexes **I** and **Ia** in the LS form, there is a small residual effective magnetic moment, which may be due to van Vleck temperature-independent paramagnetism. For compound **I**, the residual magnetic moment is 0.3 μ_β_; for **Ia** it is 0.6 μ_β_. In the cases of **IIa** and **IIIa**, as well as complex **II**, the lowest values at the temperature of liquid nitrogen are 3.92, 3.43 and 3.94 μ_β_, respectively. 

Complex **III** is in HS form throughout the studied temperature range (Figure 6). The linear dependence of the inverse magnetic susceptibility 1/(χ′) on temperature made it possible to make an approximation in the form of the Curie-Weiss law (Figure 6 shows it as a solid line):χ′(T) = N_A_·µ_β_^2^·µ_eff_^2^/(3k·(T−θ)),
here T is the temperature, k is the Boltzmann constant, N_A_ is the Avogadro constant, µ_β_ is the Bohr Magneton, µ_eff_ is the effective magnetic moment and θ is the Weiss constant [21,22].

It should be noted that in Figure 4 and Figure 5, the effective magnetic moment as a function of temperature was calculated using the formula μ_eff_ = (8χ′_M_T)^1/2^; and it is a macroscopic quantity, generally indicating the interaction between paramagnetic centers. In the form of the Curie-Weiss law, the µ_eff_ is a microscopic quantity corresponding to the effective magnetic moment of the paramagnetic center. In case of the absence of interaction between paramagnetic centers (an ideal paramagnetic) or at temperatures at which this interaction can be neglected (T >> θ), these values are the same. Approximation of experimental data in the form of the Curie-Weiss law made it possible to establish the antiferromagnetic nature of the exchange interaction (θ = −15 K) for complex **IIIa** and to obtain the value of μ_eff_ = 4.80 μ_β_ for the Fe(II) ion.

The study of the temperature dependences of the second μ_eff_(T) derivative (Figure 4 and Figure 5) allowed us to determine the temperatures of the direct (T_c_↑) and reverse (T_c_↓) transitions for the studied compounds (Table 5). For complex **Ia**, a one-stage transition with a small hysteresis is observed (T_c_↓ = 213 K, T_c_↑ = 217 K, ΔT_c_ = 4 K). In the case of the initial compound **I**, the nature of the μ_eff_(T) dependence is more intricate and the transition is two-stage. It should be noted that the hysteresis value for the low-temperature stage is the same as for the dehydrated complex (T_c1_↓ = 163 K, T_c1_↑ = 167 K, ΔT_c1_ = 4 K). At the same time, the hysteresis value for the second stage is 2.5 times greater (T_c2_↓ = 197 K, T_c2_↑ = 207 K, ΔT_c2_ = 10 K) than for the low-temperature stage. A similar nature of the effect of crystallization water can be observed in the cases of complexes **II** and **IIa**, for which hysteresis for the low-temperature stage is not observed. (T_c1_↓ = T_c1_↑ = 112 K, ΔT_c1_ = 0 K). The second stages for complexes **II** and **IIa** demonstrate significant hysteresis (**II**: T_c2_↓ = 204 K, T_c2_↑ = 234 K, ΔT_c2_ = 30 K; **IIa**: T_c2_↓ = 201 K, T_c2_↑ = 222 K, ΔT_c2_ = 21 K). In the presence of crystallization water, the hysteresis value is 1.4 times greater than in the dehydrated state. In the case of complexes **III** and **IIIa** in the studied temperature range, SCO is observed only for **IIIa**, and the transition is one-stage (T_c_↓ = T_c_↑ = 121 K, ΔT_c_ = 0 K).

## 3. Discussion

### 3.1. Structure of the [CuLCl_2_]

Pentacoordinated Cu(II) complex **IV** was isolated with one tridentate chelate ligand and two chloro ligands, demonstrating square pyramidal geometry with CuN_3_Cl_2_. The lengths of the N–Cu(II) bonds vary depending on the coordinated heterocycle; the shortest ones are observed for N atoms of the pyridine moiety, while two Cu–N(imidazole) bonds are noticeably longer (Table 2). This trend is observed for the compounds featuring similar coordination nodes [23]. According to the authors of the article [24], the bond lengths of N(4R-pyridine)–Cu(II) vary depending on the fragment in the 4th position of the ring, and the introduction of an electron-donating group into the 4-position of pyridine led to a decrease in the Cu–N bond length.

Neighboring molecules are located with an offset in the equatorial plane, so, the structure is stabilized by stacking interactions between the two nearly planar organic ligands and pseudo-dimerization is observed. It should be noted that the possibility of formation such stacking interactions between ligand parts is very characteristic of five-coordinated copper(II) complexes. Usually, this coordinate in the complex is supplemented by long-distance contacts with small ligands, but in this case we can see the organic ligand, which is sufficiently large and rigid, and apparently stacking additional benefits in energy.

### 3.2. IR Spectra of the Complexes ***I**–**III***

In the high-frequency region of IR spectra **I**–**III**, bands with a maximum at 3500–3365 cm^−1^, associated with ν(OH) vibrations are observed; they indicate the presence of crystallization water in the composition of complexes. In the IR spectra of both the ligand and all complexes in the region of 3160–3110 cm^−1^, there are bands associated with ν(NH) vibrations, and for the complexes the bands are slightly shifted to the high frequency range, thus they indicate the formation of hydrogen bonds of the network. Within the range of 3090–2830 cm^−1^ of the **L** and **I**–**III** spectra, there are bands of ν(CH) vibrations of pyridine and imidazole rings and methyl group of **L**, whose positions in all spectra coincide well with each other. In the spectra of both the ligand and all complexes in the region of 2815–2800 cm^−1^, bands associated with ν(O–CH_3_) oscillations are observed, and for complexes they are slightly shifted to the low-frequency range.

Bands of valence and deformation oscillations of heterocycles are observed in the IR spectrum **L** in the region of 1610–1430 cm^−1^, which are very sensitive to coordination. In spectra **I**–**III**, these bands appear within the range of 1680–1400 cm^−1^. A significant change in the range indicates coordination of N atoms of imidazole and pyridine to Fe(II) [25]. 

It should be noted that the spectra of complexes **I**–**III** practically coincide not only in the region of ν(C–H), but also in the region of skeletal oscillations within the range of 1400–700 cm^−1^. This confirms the conclusion about the isotype of the cations of the studied compounds.

In spectra **I**, **II**, there are bands of valence vibrations of the corresponding anions that are not split and are not displaced in comparison with the bands positions in the IR spectra of the potassium salts of these anions, which indicate their external position.

### 3.3. DRS of the Complexes ***I**–**III***

From the DRS data, it is easy to calculate the values of the ligand field splitting parameter in the crystal of high-spin Fe(II) complexes Δ_HS_ = ν(^5^T_2_ → ^5^E) and Racah parameters **B** = Δ_HS_/19; **C** = 4.41 B (Table 6). The average electron pairing energy (**P**) for high-spin iron(II) ions was approximately calculated based on the spectral data obtained by the formula: P = ν_1_ + Δ_HS_ − ν_2_ [26]. The resulting values are slightly higher than the value P = 17,600 cm^−1^ for [Fe(H_2_O)_6_]^2+^ [27].

The reviews [28,29,30] define the conditions for the SCO’s existence as follows: ${{∆}_{HS}=10Dq}^{HS}<P<{10Dq}^{LS}={∆}_{LS}$. If 10Dq^HS^ < 10,000 cm^−1^, the complex remains high-spin at all temperatures; if 10Dq^HS^ ≅ 11,000–12,500 cm^−1^, the complex undergoes SCO when cooled. The implementation of the first inequality and the approximation to a non-strict equality for the value of the splitting parameters indicate that the SCO manifestation in complexes **I**–**III** is very likely.

### 3.4. Mössbauer Spectra of the Complexes ***I**–**III***

Mössbauer spectra of previously studied iron(II) complexes with 2,6-bis(1*H*-imidazol-2-yl)pyridine derivatives [10,13,15] with the same anions as in this work show that in all these compounds Fe(II) is only in the LS state at room temperature, with the exception of the iron complex with 2,6-bis(4,5-dimethylimidazol-2-yl)pyridine (**L***) of the composition [Fe(L*)_2_]SO_4_·0.5H_2_O [13]. On the contrary, the spectra of the samples studied in this work indicate the presence of Fe(II) only in the HS state. A comparison of the spectra of the complexes [FeL_2_]SO_4_∙2H_2_O (**I**) and [Fe(L*)_2_]SO_4_·0.5H_2_O shows that they have almost the same isomeric shift, but significantly different values of quadrupole splitting (1.88 mm/s compared to 2.25 mm/s). This distinction may be due to a difference in the structure of ligands and/or the content of hydrated water molecules, as well as to a different distortion of the coordination polyhedron FeN_6_; these factors can lead to a different arrangement of ions in the crystal lattice, which in turn will lead to a change in the gradient of the electric field on the iron core.

### 3.5. Magnetic Properties of the Complexes

As shown above, the studied compounds exhibit spin crossover at temperatures below room temperature. It should be noted that when the auxochromic methoxy group was introduced into the composition of 2,6-bis(1*H*-imidazol-2-yl)pyridine, for complexes with 2,6-bis(1*H*-imidazol-2-yl)-4-methoxypyridine, the spin crossover temperatures significantly shifted to the low temperature region [14,15]. 

It should also be noted that the nature of the spin crossover is significantly influenced by crystallization water. First of all, for complex **I**, the presence of crystallization water complicates the form of μ_eff_(T) dependence; thus, the transition for **I** is two–stage, while for **Ia** is one-stage. For the studied compounds, the presence of crystallization water leads to an increase in the μ_eff_ in the high-spin state and may cause a decrease in the residual μ_eff_ in the low-spin state. It is important to note that in these compounds, SCO is observed in the cryogenic region close to the lower limit of the temperature range of the Faraday balance (boiling point of liquid nitrogen). In this regard, it is impossible to reliably conclude whether the observed low-temperature values of µ_eff_ for complexes **II**, **IIa** and **IIIa** are due to the residual magnetic moment or that the SCO continues at temperatures below the temperature range of the device. As shown earlier [10,11,12,13,14,15,16,17], crystallization water has a significant effect on the transition temperature. For the pair **I** and **Ia**, the presence of crystallization water lowers the transition temperature of the first stage. In the case of **II** and **IIa**, the temperature of the first stage does not change, but there is a trend to decrease the temperatures of the direct and reverse transitions for the second stage. For **III** and **IIIa,** it was shown that the crossover is not observed for the initial complex in the studied temperature range. At the same time, it cannot be reliably stated that for the **III** complex the crossover is not observed in the temperature range below the boiling point of liquid nitrogen. Thus, it can be concluded that for the studied compounds, the presence of crystallization water may cause a decrease in the temperatures of the direct and reverse SCO for at least one of the crossover stages. Crystallization water also has a significant effect on the hysteresis value (Table 5). So, in the case of the pair **I** and **Ia**, the crystallization water causes the appearance of the second stage of the transition, the hysteresis for which is 2.5 times greater than for the first stage. For a pair **II** and **IIa**, the presence of crystallization water causes a hysteresis increase by 1.4 times.

Thus, the crystallization water for the studied compounds influences both the values of the effective magnetic moments and the SCO temperature as well as the hysteresis values.

## 4. Materials and Methods

All commercially available solvents and reagents were analytical grade and used without further purification. 

FeSO_4_·7H_2_O from Acros Organics, KReO_4_ and BaBr_2_ from Sigma-Aldrich, ascorbic acid of the “medical” qualification, and ethanol “rectificate” were used for the synthesis of complexes.

**Dimethyl 4-hydroxypyridine-2,6-dicarboxylate (1) and dimethyl 4-methoxypyridine-2,6-dicarboxylate (2)**. Chelidamic acid hydrate (20 g, 99.5 mmol) was dissolved in a solution of MeOH (150 mL) and concentrated H_2_SO_4_ (17 mL). The mixture was refluxed for 24 h and then allowed to cool down to room temperature. A saturated NaHCO_3_ solution was added up to pH 2.0, the residue was filtered off, washed with H_2_O and dried to give **1** as a white solid (7.4 g, 35%). The filtrate was extracted with CH_2_Cl_2_, the combined organic extracts were dried over MgSO_4_ and the solvent was removed under vacuum to give **2** as a white solid (7.5 g, 34%).

**(1)**: m.p. 110 °C (decomp.). IR spectrum (KBr), ν, cm^−1^: 3433, 1741. ^1^H NMR spectrum (400 MHz), δ, ppm: 3.84 s (6H, 2CH_3_OC(O)), 7.52 s (2H, H-3, H-5). ^13^C NMR spectrum, (100 MHz), δC, ppm: 166.4, 165.2, 149.6, 115.7, 53.1. Mass spectrum: *m*/*z* calcd. for C_9_H_9_NO_5_ [M]^+^ 211.0475; found 211.0478.

**Dimethyl 4-methoxypyridine-2,6-dicarboxylate (2)**. Dimethyl 4-hydroxypyridine-2,6-dicarboxylate **1** (6.8 g, 32.2 mmol) was heated with CH_3_I (5.5 g, 2.4 mL, 38.7 mmol) and K_2_CO_3_ (6.7 g, 48.5 mmol) in DMF (50 mL) under stirring at 50 °C for 8.5 h. Then the mixture was allowed to cool down to room temperature and H_2_O (100 mL) was added. The residue was filtered off, washed with H_2_O and dried to give **2** as a white solid (5.1 g, 70%).

**(2)**: m.p. 127.8–128.2 °C. IR spectrum (KBr), ν, cm^−1^: 1716, 1726. ^1^H NMR spectrum (300 MHz), δ, ppm: 3.90 s (6H, 2CH_3_OC(O)), 3.97 s (3H, CH_3_O), 7.72 s (2H, H-3, H-5). ^13^C NMR spectrum, (125 MHz), δC, ppm: 167.0, 164.6, 149.4, 113.8, 56.2, 52.6. Mass spectrum: *m*/*z* calcd. for C_10_H_11_NO_5_ [M]^+^ 225.0632; found 225.0635.

**4-Methoxypyridine-2,6-dicarboxamide (3)**. Dimethyl 4-methoxypyridine-2,6-dicarboxylate **2** (5.1 g, 22.7 mmol) was dissolved in MeOH (190 mL) and NH_4_OH (25%, 50 mL) was added. The mixture was refluxed for 48 h. After cooling down to room temperature, the residue was filtered off, washed with Et_2_O and dried to give **3** as a white solid (4.2 g, 95%).

**(3)**: m.p. 299.7–300.0 °C. IR spectrum (KBr), ν, cm^−1^: 3317, 3288, 1672. ^1^H NMR spectrum (300 MHz), δ, ppm: 4.01 s (3H, CH_3_O), 7.80 s (2H, H-3, H-5). ^13^C NMR spectrum, (125 MHz), δC, ppm: 168.5, 167.9, 150.0, 111.3, 56.1. Mass spectrum: *m*/*z* calcd. for C_8_H_9_N_3_O_3_ [M]^+^ 195.0638; found 195.0634.

**4-Methoxypyridine-2,6-dicarbonitrile (4)**. 4-Methoxypyridine-2,6-dicarboxamide **3** (3.5 g, 17.95 mmol) was suspended in THF (25 mL) under stirring at 0 °C and Et_3_N (9.1 g, 90.0 mmol) was added. Trifluoroacetic anhydride (9.4 g, 44.76 mmol) was slowly added by dropwise for 1 h. Then the mixture warmed to room temperature and stirred overnight. A saturated NaHCO_3_ solution was slowly added to pH 8.0, the residue was filtered off, washed with H_2_O, dried and washed with hexane, dried in vacuo to give **4** as a white solid (1.88 g, 66%).

**(4)**: m.p. 137–139 °C. IR spectrum (KBr), ν, cm^−1^: 2243, 1591. ^1^H NMR spectrum (400 MHz), δ, ppm: 3.97 s (3H, CH_3_O), 7.37 s (2H, H-3, H-5). ^13^C NMR spectrum, (125 MHz), δC, ppm: 166.8, 136.2, 117.5, 115.4, 56.5. Mass spectrum: *m*/*z* calcd. for C_8_H_5_N_3_O [M]^+^ 159.0427; found 159.0429. 

**2,6-Bis(1*H*-imidazol-2-yl)-4-methoxypyridine (L)**. 4-Methoxypyridine-2,6-dicarbonitrile **4** (1.08 g, 6.79 mmol) was dissolved in MeOH (6.8 mL) and sodium methoxide (0.3 mL of 30% solution in MeOH) was added. The mixture was stirred for 2.5 h at rt and then aminoacetaldehyde diethyl acetal (1.8 g, 13.5 mmol) was added. The reaction mixture was acidified with acetic acid (0.8 mL), heated to 50 °C for 1 h and then cooled to rt. Then MeOH (14 mL) and 6 N HCl in H_2_O (3.4 mL) were added, and the reaction mixture was heated to reflux for 5 h. After cooling down to room temperature, the obtained solid was diluted with MeOH (5 mL), filtered off, and washed with MeOH (5 mL). The residue was washed with a solution of NaOH (1 g) in H_2_O (12 mL) and H_2_O adjusted to pH 8.0, then dried under vacuum to give **L∙1.2H_2_O** as a white solid (1.2 g, 67%).

**(L∙1.2H_2_O)**: m.p. 268°C (decomp.). IR spectrum (KBr), ν, cm^−1^: 3383, 3138, 3113, 1606, 1570, 1551, 1477, 1437. ^1^H NMR spectrum (400 MHz), δ, ppm: 3.95 s (3H, CH_3_O), 7.13 s (2H, H_Ar_), 7.39 s (2H, H_Ar_). 7.47 s (2H, H_Ar_), 12.65 s (2H, NH). ^13^C NMR spectrum, (100 MHz), δC, ppm: 167.6, 149.9, 146.0, 130.22, 130.18, 119.0, 118.9, 103.6, 56.1. Mass spectrum: *m*/*z* calcd. for C_12_H_11_N_5_O [M]+ 241.0958; found 241.0959. 

Found: % C = 54,7; % H = 4.9; % N = 26.4.

Anal. Calc. for C_12_H_11_N_5_O∙1.2H_2_O (262.87 g/mol): % C = 54.8; % H = 5.1; % N = 26.6.

**Synthesis of [FeL_2_]SO_4_∙2H_2_O (I)**. A weighted portion of FeSO_4_·7H_2_O (0.14 g, 0.5 mmol) was dissolved in 3 mL of distilled water acidified with 0.05 g ascorbic acid. A solution of **L∙1.2H_2_O** (0.26 g, 1 mmol) in 10 mL of ethanol was slowly added to the resulting solution. After mixing the solutions, a bright orange solution was formed, from which a fine orange precipitate fell out within a few minutes. The precipitate was kept in solution while being stirred on a magnetic stirrer for 3 h. Precipitate **I** was filtered out, washed several times with water and ethanol, and dried in the air (0.22 g, 73%).

Found for **I**: % C = 44.4; % H = 4.1; % N = 20.1; % S = 4.5. Anal. Calc. for C_24_H_26_N_10_O_8_FeS (670.44 g/mol): % C = 43,0; % H = 3.9; % N = 20.9; % S = 4.8.

**Synthesis of [FeL_2_](ReO_4_)_2_∙H_2_O (II)**. Synthesis was carried out similarly to **I**, but at the stage of formation of a bright orange solution, without waiting for **I** precipitation, a hot KReO_4_ solution (0.29 g, 1 mmol in 10 mL H_2_O) was added. A fine orange precipitate instantly fell out, which was kept in solution while being stirred on a magnetic stirrer for 1 h. Precipitate **II** was filtered out, washed several times with water and ethanol, and dried in the air (0.45 g, 95%).

Found for **II**: % C = 29.0; % H = 2.5; % N = 13.9. Anal. Calc. for C_24_H_23_FeN_10_O_10.5_Re_2_ (1056.78 g/mol): % C = 27.5; % H = 2.2; % N = 13.4.

**Synthesis of [FeL_2_]Br_2_∙2H_2_O (III)**. A weighted portion of FeSO_4_·7H_2_O (0.14 g, 0.5 mmol) was dissolved in 3 mL of distilled water acidified with 0.05 g ascorbic acid. A warm solution of BaBr_2_ (0.15 g, 0.5 mmol in 3 mL H_2_O) was slowly added to the resulting solution, and 3 h after the complete sedimentation of BaSO_4_, the solution was filtered out. A solution of **L∙1.2H_2_O** (0.26 g, 1 mmol in 10 mL of ethanol) was added to the mother liquor. After mixing, a dark orange solution was formed, from which a fine orange precipitate fell out within an hour. Precipitate **III** was filtered out, and washed several times with water and ethanol, dried in air (0.30 g, 91%).

Found for **III**: % C = 39.7; % H = 3.9; % N = 19.2. Anal. Calc. for C_24_H_26_FeN_10_O_4_Br_2_ (734.19 g/mol): % C = 39.3; % H = 3.6; % N = 19.1. 

**Synthesis of [CuLCl_2_] (IV)**. CuCl_2_·2H_2_O (0.0085 g, 0.05 mmol) was dissolved in 2 mL of distilled water acidified with 0.005 g ascorbic acid. A solution of **L∙1.2H_2_O** (0.026 g, 0.1 mol in 5 mL of ethanol) was slowly added to the resulting solution. After mixing, a green solution was formed, from which green crystals fell out within a week. Crystals **IV** were filtered out, washed several times with water and ethanol, and dried in the air (0.015 g, 79%).

Found for **IV**: % C = 37.8; % H = 3.1; % N = 18.1. Anal. Calc. for C_12_H_11_N_5_OCuCl_2_ (375.70 g/mol): % C = 38.4; % H = 3.0; % N = 18.6. 

**The IR spectra** of organic compounds were recorded in KBr on a Bruker Vector-22 spectrometer. 

**The ^1^H and ^13^C NMR spectra** were recorded on Bruker AV-300 (300.13 and 75.5 MHz, respectively), Bruker AV-400 (400.13 and 100.61 MHz), Bruker DRX-500 (500.13 and 125.76 MHz) spectrometers (Germany) at room temperature using as solvents CDCl_3_, DMSO-d_6_, D_2_O (purity 99.8%); the chemical shifts were measured relative to the residual proton and carbon signals of the deuterated solvent with respect to TMS as the internal standard. 

**The melting points** were measured with a Mettler Toledo FD-900 melting point apparatus. The high-resolution mass spectra (electron impact, 70 eV) were run on a Thermo Electron DFS instrument. 

**The progress of reactions** and the purity of the isolated compounds were monitored by TLC on Sorbfil PTLC-AF-A-UV plates using chemically pure grade chloroform as eluent. 

**The CHNS elemental analyses** of complexes were performed on a EuroVector EA3000 Elemental Analyser.

**Thermal analysis** of the samples was performed on an STA 409 PC Luxx simulta-neous thermal analyzer manufactured by NETZSCH-Gerätebau GmbH. Thermogravimetric (**TG**) and differential scanning calorimetric (**DSC**) data were recorded during the experiment. The analysis was carried out in corundum ceramic crucibles. The heating was carried out at a rate of 10 K/min in the air.

**X-ray powder diffraction** data were collected on a Shimadzu XRD 7000 diffractometer (CuK_α_ radiation (λ = 1.5406 Å), Ni filter, scintillation detector) at room temperature.

**Single-crystal X-ray diffraction** data were collected using the graphite monochromatized MoK_α_-radiation (λ = 0.71073 Å) at 150(2) K on the X8APEX Bruker Nonius diffractometer equipped with a 4K CCD area detector. The ϕ-scan technique was employed to measure intensities. Absorption corrections were applied empirically using the SADABS program [31]. Structure was solved by the direct methods of the difference Fourier synthesis and further refined by the full-matrix least squares method using the SHELXTL package [32]. Atomic thermal parameters for non-hydrogen atoms were refined anisotropically. The positions of hydrogen atoms were calculated corresponding to their geometrical conditions and refined using the riding model. The main parameters of structural experiments are listed in Table 1. The coordinates of atoms and other parameters for structure **IV** were deposited with the Cambridge Crystallographic Data Centre (**CCDC 2252412**, www.ccdc.cam.ac.uk/data_request.cif accessed on 29 March 2023). Selected bond lengths and angles are listed in Table 2.

**IR absorption spectra** were registered on IRAffinity-1S (Shimadzu) and Scimitar FTS 2000 spectrometers within the range of 4000 to 400 cm^−1^, and Vertex 80 spectrometers within the range of 400 to 200 cm^−1^. The samples were prepared in the form of suspensions in KBr, vaseline and fluorinated oils as well as in polyethylene.

**The diffuse reflectance spectra (DRS)** were recorded on a Shimadzu UV-3101 PC scanning spectrometer at room temperature.

**The Mössbauer spectra** of iron(II) complexes were measured at a room temperature using an NP-610 spectrometer with a ^57^Co (Rh) radiation source. The spectra were processed to determine the values of isomer shift δ (with respect to α-Fe) and quadrupole splitting ΔE_Q_.

**Static magnetic susceptibility** was measured by the Faraday method using torsion quartz microbalance with electromagnetic compensation. The temperature stabilization of the samples (~1 K) in the temperature range 80–360 K was carried out with the Delta DTB9696 temperature controller (Delta Electronics, China). The heating or cooling mean rate was ~2–3 K/min. The fluctuations in magnetic field strength (7300 Oe) were less than 1%. The study of dehydrated complexes (**Ia**–**IIIa**) was carried out after the vacuum drying of the samples in the measurements chamber. The samples (~20 mg) placed in the open quartz cellules were vacuumed at a pressure 10^−2^ torr. After that, the chamber was filled up with the helium at the pressure 5 torr. The investigated compounds **I**–**III** were placed in sealed quartz cellules with the air atmosphere at 760 Torr. The values of the effective magnetic moment were calculated after correction for the diamagnetic contribution according to the Pascal scheme as μ_eff_ = (8χ′_M_T)^1/2^, where χ′_M_ is corrected molar magnetic susceptibility. The temperatures of the direct (T_c_↑) and reverse (T_c_↓) transitions were determined using the condition d^2^(μ_eff_)/dT^2^ = 0.

## 5. Conclusions

In this work, we synthesized new compounds of iron(II) with 2,6-bis(1*H*-imidazol-2-yl)-4-methoxypyridine (**L**) of the compositions [FeL_2_]SO_4_∙2H_2_O, [FeL_2_](ReO_4_)_2_∙H_2_O and [FeL_2_]Br_2_∙2H_2_O, as well as the copper(II) complex [CuLCl_2_]. To do this, we have presented a convenient synthetic pathway for the synthesis of an imidazole-pyridine-based ligand with an auxochromic group in the central moiety. The study of the temperature dependence of μ_eff_ of the obtained complexes showed that they exhibit the ^1^A_1_ ↔ ^5^T_2_ spin-crossover, the nature and temperature of which depend on the composition of the compound. The synthesized complexes are bifunctional, the spin-crossover in them is accompanied by thermochromism (reversible color change from orange to red-violet), which is of independent interest.

## Data Availability

The data presented in this study are available on request from the corresponding authors.

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
