# Peer review of "Spin Crossover and Thermochromism in Iron(II) Complexes with 2,6-Bis(1H-imidazol-2-yl)-4-methoxypyridine"

_ijms, 2023, doi:10.3390/ijms24129853_

Round 1

Reviewer 1 Report (Previous Reviewer 2)

I propose to accept the work in the currently sent form.

Author Response

We thank the reviewer for the high assessment of our work.

On behalf of the authors, with respect,

Guest Editor of Special Issue IJMS

Prof. Dr. Lyudmila G. Lavrenova

and Prof. Dr. Olga G. Shakirova

Reviewer 2 Report (New Reviewer)

The paper of Lyudmila G. Lavrenova and co-authors is an interesting fundamental work on synthesis iron(II) compounds with 2,6-bis(1H-imidazol-2-yl)-4-methoxypyridine and investigation of their magnetic properties. The authors confirmed the presence of iron(II) using Mössbauer spectroscopy and detected the behavior of the spin-crossover with a change in the magnetic moment of the complexes.

Major

When studying magnetic behavior, it is very important to confirm the phase purity of compounds. The authors presented phase diagrams (Figure S15. Diffractograms of complexes). Experimental powder diffractogram seems clear, do not contain amorphous phase. But without additional analysis, it is impossible to draw a conclusion about the composition: for example, it can be a stoichiometric mixture of the starting reagents. It would be better to grow a single crystal, compare for exclusion of any doubts diffractograms by Rietveld method using the TOPAS program with the single crystal data as a model. Unfortunately, the authors failed to grow single crystals of the iron(II) compounds.

So, there is a lack of research on the composition of complexes I-III. The elemental analysis data may coincide, for example, with a stoichiometric mixture of the ligand and the starting iron salts. I suggest that the authors do a mass spectrometric analysis to detect the molecular ion of complexes I-III. Electrospray ionization is best, but MALDI will also be sufficient.

Without additional research on the composition of complexes I-III, I could recommend tranferring this article from IJMS to a more appropriate journals - Magnetochemistry or Inorganics.

Minor

Scheme 1, which describes the ligand synthesis, does not contain reaction yields. At least the output of the last stage should be added (in comparison with the original amount)

In my humble opinion it was not the best decision to describe the copper complex with a ligand in classic spin-crossover article. It seems to me, at the discretion of the authors, this part can be removed and published separately in such journals as Molbank or Journal of Structural Chemistry.

Author Response

We agree with the distinguished reviewer that the data of the elemental analysis for the complex in some cases may coincide with the data for the stoichiometric mixture of the ligand and the initial iron salts. However, in our case, the formation of the complex is reliably confirmed by a number of physico-chemical methods.

  1. By the method of static magnetic susceptibility, it is shown that the complexes exhibit a spin crossover of 1А1 « 5Т2. As is known, the Fe(II) ion has 6 electrons in d-orbitals. During the formation of the complex in the crystal field of ligands, splitting into two sublevels is observed: t2g and eg. In this case, there can be two states. If all six d-electrons are paired and located at the sublevel t2g, the total spin S=0, multiplicity 2S+1=1, the state 1A1 is realized. Under the influence of external conditions (see the introduction to the article), 4 electrons are steamed, two of them are located at the sublevel t2g and two at eg. In this case, the total S=2, multiplicity 2S+1=5, the state 5T2 is realized. This is the spin crossover, and this phenomenon cannot be observed for pure iron(II) ion.
  2. The Mossbauer spectra show that at room temperature the complexes are in a high spin state, and this clearly indicates their existence.
  3. In the diffuse reflection spectra, bands of d-d transitions are observed in the visible region, and such transitions cannot manifest themselves in the ligand spectrum (ligand bands are in the UV region).
  4. In the IR spectra, a shift of ligand bands is observed, which can only be observed during the formation of complexes. This is discussed in detail in the article.
  5. We can also mention the color change that accompanies complexation. The iron(II) ion has a light green color, and the ligand is colorless. As a result of complex formation, an orange complex is formed, which has thermochromism: when cooled, its color turns into red-purple. Thermochromism in this case accompanies the spin crossover.

Thus, we have confirmation of the formation of the complex by a number of methods, and therefore we do not see the need to use another method to confirm this indisputable fact. 

This article was prepared for a special issue of IJMS “The Design, Synthesis and Study of Metal Complexes”.

We thank the reviewer for comment and have entered the output data into scheme 1. The reaction yields in comparison with the original amount of chelidamic acid hydrate given in brackets.

We agree with the reviewer that the X-Ray data for the obtained iron(II) complexes would be more appropriate in the article. However, it is not always possible to obtain single crystals for iron(II) complexes due to the fact that the Fe2+ ion is unstable. In this case, it is necessary to rely in the conclusions about the structure on the X-Ray data for complexes of other 3d metals with the same ligand. Complexes with 2,6-bis(1H-imidazol-2-yl)-4-methoxypyridine were obtained for the first time, so it was important for us to know the method of ligand coordination according to X-Ray data. And the use of a number of other physicochemical methods allowed us to conclude that the method of ligand coordination coincides in copper(II) and iron(II) complexes.

On behalf of the authors, with respect,

Guest Editor of Special Issue IJMS

Prof. Dr. Lyudmila G. Lavrenova

and Prof. Dr. Olga G. Shakirova

Round 2

Reviewer 2 Report (New Reviewer)

Dear authors,

Thanks for such a complete answer. Indeed, indirect methods confirm the formation of iron compounds. It would be nice to see in future papers mass spectra of similar compounds for which it is impossible to grow crystals. You can also use paramagnetic NMR, for iron compounds this can be done. I can also suggest the Evans method for measuring the change in the magnetic moment - this already old method can still give important results.

I suggest that this work be accepted as is, perhaps it will cause discussion among inorganic chemists, which in any case will lead to a citation of the article and an increase in the impact factor of the journal.

Sincerely

This manuscript is a resubmission of an earlier submission. The following is a list of the peer review reports and author responses from that submission.

Round 1

Reviewer 1 Report

The manuscript by Shakirova et al describes a synthetic pathway to the ligand 2,6-bis(1H-imidazol-2-yl)4-methoxypyridine (already described in literature) and its copper(II) and three iron(II) complexes with different counter ions. The copper complex was characterized via single crystal X-ray structure analysis, while for the iron complexes the magnetic properties were characterized. It is a bit unusual, as there are no structural similarities to be expected between the copper complex and the iron complexes, starting with the expected difference in ligands coordinated to the metal center. Furthermore, there are a number of open questions:

1.       Please explain the motivation for the choice of the ligand and in the case of the iron complexes for the choice of the counter ions, that are rather unusual.

2.       The term X-ray phase analysis is unusual, only the diffraction patterns are shown but not further analysed. What do you mean by CuLCl2 phase?

3.       There are a wide number of groups that investigate iron complexes with pyridine-imidazole-based ligands and their switching properties. Please give a proper and complete literature overview!

4.       Please explain why it is important to determine the van Vleck paramagnetism of iron(II) complexes.

5.       The quadrupole splitting is usually given as DeltaEQ. Please explain why it is different for II compared to I and III.

6.       Please use always the same scale when comparing the magnetic properties of the complexes.

7.       It is unusual to have the discussion separated from the experimental results but with the same substructure. This leads to duplication and unnecessary lengthening of the manuscript and should be avoided. Furthermore, the discussion of the SCO/Mossbauer properties without of any structural data remains highly speculative.  The authors need to provide some X-ray structures to allow any discussion here. Furthermore, it would be helpful to compare hydrated/dehydrated samples not only by magnetic measurements but also using the other methods.

At the present stage, the manuscript is incomplete and not suitable for publication.

in part unusual abbreviations and terms are used

Reviewer 2 Report

The authors synthesised novel ligand 2,6-bis(1H-imidazol-2-yl)-pyridine by an incorporation of methoxy group into 4 position of pyridine ring. It succesfully allowed to prepare three novel Fe(II) complexes and one Cu(II) mononuclear system which was also characterized by SCXRD studies. The Fe(II) complexes were detailed studied which allowed to confirm occurence of thermally induced spin crossover. The manuscript should be publicate  however improvements  are required.

1.      In my opinion presentation of results of magnetic studies in the form of xT vs T (v = molar magnetic susceptibility, T  = temperature) will be more convinient for readers because there are presented two step or/and incomplete spin crossover and hence xT(T) dependence  is allow to fast estimate HS fraction.

2.      Results of elemental analyse for novel ligand should be added.

3.      Authors stated that „All compounds are stable for a long time when stored in the air at room temperature.” According to results of magnetic studies the compound I (freshly prepared) can exhibit two step spin crossover (as inherent feature of hydrated form) however it is also probable an existence of mixture of solvated and desolvated forms (each undergoing spin crossover at different temperature). This second assuption seems to support results of elementl analysis where experimentally determined carbon content is higher then the calculated for hydrated form (some discrepancy for nitrogen in compound I appears) . Authors could be considered addition of results of elemental analyses for desolvated samples.  In may opinion it is important because magnetic studies  were carried out for hydrated samples placed in sealed tubes as well as for samples placed in open tubes where dehydration most probably occured (0.01 torr) however it was not confirmed analytically.  Authors should discuss this question in details.

4.      The authors were able to determin the crystal structure for Cu(II) compound, however, there is some discrepancy of results of elementary analysis. It should be explained.

5.      Moessbauer spectroscopy measurements were done for samples I, II and III. Each spectrum is composed from one component  characteristic for HS form. Taking into account results of magnetic studies, in particular for compound I, it should be expected the presence of second component.  What is the reason the only one component is present on the spectrum of compound I?

6.      Authors were not able to determine the crystal structures I-III. Nevertheless results of elemental analysis remain in reasonable agreement with composition [Fe(L2]2+ . Hence it seems that metal centers could be rather isolated and hence  it could be expected only very week magnetic interactions.  What could be reasons that inverse molar magnetic susceptibility vs temperature leads to quite large negative theta value for III??

7. Transition 5T2 to 5E is usually located at ca 850 cm-1 and is very weak. Authors should give in experimental section details concerning collection of DRS spectra (standard references, line corraction) and subsequent spectra deconvolution.

Round 2

Reviewer 2 Report

In my opinion the manuscript can be published in the present form.